

# Microbial soil characteristics of grassland and arable soils linked to thermogravimetry data: correlations, use and limits

Helena Doležalová-Weissmannová[1], Stanislav Malý[2], Martin Brtnický[1, 3], Jiří Holátko[3], Michael Scott Demyan[4], Christian Siewert[5, 6], David Tokarski[6, 7], Eliška Kameníková[1], and Jiří Kučerík[1]

[1]Institute of Chemistry and Technology of Environmental Protection, Faculty of Chemistry, Brno University of Technology, Purkyňova 118, 61200 Brno, Czech Republic
[2]Central Institute for Supervising and Testing in Agriculture, Hroznová 2, 65606 Brno, Czech Republic
[3]Department of Agrochemistry, Soil Science, Microbiology and Plant Nutrition, Faculty of AgriSciences, Mendel University in Brno, Zemědělská 1, 61300 Brno, Czech Republic
[4]School of Environment and Natural Resources, The Ohio State University, 2021 Coffey Road, Columbus, OH 43210, USA
[5]Technical University Berlin, Institute of Ecology, Chair of Soil Conservation, Ernst-Reuter-Platz 1, D-10587 Berlin, Germany
[6]Technische Universität Dresden, Faculty of Environmental Sciences, Institute of Soil Science and Site Ecology Pienner Straße 19, Tharandt 01737, Germany
[7]LKS - Landwirtschaftliche Kommunikations-und Servicegesellschaft mbH, August-Bebel-Straße 6, 09577 Niederwiesa, Germany

**Correspondence:** Jiří Kučerík (kucerik@fch.vut.cz)

**Abstract.** Thermogravimetry (TG) is a simple method that enables rapid analysis of soil properties such as content of total organic C, nitrogen, clay and C fractions with different stability. However, the possible link between TG data and microbiological soil properties has not been systematically tested yet and limits TG application for soil and soil organic matter assessment. This work aimed to search and to validate relationships of thermal mass losses (TML) to total C and N contents, microbial biomass
C and N, basal and substrate-induced respiration, extractable organic carbon content, anaerobic ammonification, urease activity, short-term nitrification activity, specific growth rate, and time to reach the maximum respiration rate for two sample sets of arable and grassland soils. Analyses of the training soil set revealed significant correlations of TML with basic soil properties such as carbon and nitrogen content with distinguishing linear regression parameters and temperatures of correlating mass losses for arable and grassland soils. In a second stage the equations of significant correlations were used for validation with an
independent second sample set. This confirmed applicability of developed equations for prediction of microbiological properties mainly for arable soils. For grassland soils was the applicability lower, which was explained as the influence of rhizosphere processes. Nevertheless, the application of TG can facilitate the understanding of changes in soil caused by microorganism's activity and the different regression equations between TG and soil parameters reflect changes in proportions between soil components caused by land use management.



## 1   Introduction

Assessment of soil quality and health is currently based on complex soil quality indexes (SQI), which require the measurement of physical, chemical and/or biological properties (Gil-Sotres et al., 2005). Currently, three groups of SQI are recognized: chemical, physical and biological. Chemical indicators include pH, total and organic C, total N, extractable P, nutrient level; among physical indicators belong soil texture, rooting depth, electrical conductivity, infiltration rate, bulk density, water reten-
tion capacity and biological SQI are represented by C and N microbial biomass, potentially mineralizable N, soil respiration, soil enzyme activities and others. Different SQI parameters have different sensitivity towards shifts in soil processes and composition. The physical parameters response mainly to drastic soil changes, whereas the biological and biochemical parameters are sensitive even to very slight soil modifications. Also chemical parameters can be in specific cases very sensitive to shifts in soil processes. Hence, the SQI should always include biological, biochemical and chemical parameters, which are sensitive
and response rapidly.

However, the analysis of a wide range of properties requires a multitude of methods and approaches that may be costly, laborious, time-consuming and not achievable in all laboratories. Besides, the increasing number of methods increases the probability of experimental errors (David et al., 2019), reduces comparability of results and applicability of calculated indexes.

An alternative approach is to use a validated technique to determine many soil properties in one simple measurement with
high reliability and reproducibility. Such an approach would save time, reduce costs and decrease the probability of experimental biases, especially in studies with large number of soil samples (Askari et al., 2015).

Among most frequently discussed methods are the spectroscopic techniques such mid- and near-infrared spectroscopies (Chang et al., 2001; Demyan, 2013; Demyan et al., 2013, 2012; Giacometti et al., 2013; Knox et al., 2015; Parikh et al., 2014; Stenberg et al., 2010; Wetterlind, 2013; Zbíral et al., 2017; Zornoza et al., 2008). A second group includes methods of
thermal analysis that provide much information on soil dynamics and microbiological processes (Harris et al., 2012; Barros et al., 2016, 2011, 2010, 2007, 2000; Herrmann et al., 2014; Plante et al., 2010, 2009), on fractionation of soil organic matter according to thermal or thermo-oxidative stability (Pospíšilová et al., 2011; Coelho et al., 2013), soil moisture (Wang et al., 2011) or on soil organic C (SOC), N and clay content (Kristl et al., 2016; Siewert, 2004, 2001; Vuong et al., 2013). Among the most common thermoanalytical method is thermogravimetry (TG) by which a sample is heated using a linear heating program
with continuously monitoring of sample mass (Rotaru et al., 2008; Rotaru and Goşa, 2009). The evaluation of results is most frequently based on dependency of recorded mass losses on temperature or mass loss derivative (De Lisi et al., 2006; Donato et al., 2010). The thermal stability of soil is determined mainly by binding energies of soil organic matter (SOM) components, thermo-oxidative stability, and their accessibility to oxygen and other factors Kucerik (2017).

Besides technical simplicity, TG has several benefits. In particular, it provides both quantitative (mass loss signal) and
qualitative (temperature signal) information (Kucerik, 2017). Furthermore, the mass losses obtained in specific temperature intervals reflect some soil properties. Unlike infrared spectroscopy, where signal intensity at certain frequencies may usually be attributed to particular chemical functional groups (n.b. for example infrared is also subject to overlapping vibrational ranges), the mass losses obtained using TG do not always have a clear meaning. This is caused by overlapping of mass





losses originating from various simultaneously occurring processes (evaporation, oxidation, chemical transformation, etc.).
Therefore, the biogeochemical meaning of mass losses of soil in some temperature areas usually needs accessory information from additional experiments or by using hyphenated techniques (Fernández et al., 2012).

To avoid evaluation problems caused by these methodological constraints, an alternative evaluation procedure was introduced. In addition to traditional peak analyses, the determination of thermal mass losses (TML) in predefined temperature intervals was used. This approach facilitates sample comparison from different origins and the search for global model in
calculations and also reduces consequences of TG measurement noise (Kučerík et al., 2018).

A number of works confirmed significant correlations between TML and various soil properties. In particular, TML correlations were observed for SOC, total nitrogen (TN) and clay contents (Siewert, 2004, 2001), soil carbonates ((Siewert, 2004, 2001), amount and rate of $CO_2$ evolution measured during basal soil respiration under laboratory conditions (Kučerík et al., 2013; Siewert et al., 2012), total SOM content (Kučerík et al., 2018) and soil organic matter fractions of vary biological
turnover rates (Tokarski et al., 2020).

In other words, TG enables estimation of physical SQI components such as texture and moisture content, chemical SQI components such as total C and N, but only a limited number of biological SQI components such as soil respiration. Indeed, the biological/biochemical properties are among the most sensitive properties (Gil-Sotres et al., 2005) responding quickly to management changes. Therefore, a rapid alternative estimating multiple soil parameters in one analysis would be beneficial.
The aim of this study is to investigate to which extend thermal mass losses by TG analyses can serve as a possible tool for rapid estimation of biological and microbiological (MB) soil properties. Previously highly significant correlations between thermal mass losses and soil properties were observed mainly for soils with very limited anthropogenic influence, while we focused our current study on the comparison of arable soils with semi-natural grassland soils without regular tilling and fertilization.

## 2 Experimental

### 2.1 Soil sampling and preparation

All soils originated form a regular monitoring program of the Central Institute for Supervising and Testing in Agriculture, Czech Republic (Poláková et al., 2017). The details on soil types and properties can be found in Supporting information, Table S1 and S2.
The first two sample sets consisted of 11 grassland and 21 arable soils and were used to search for interrelations between thermal mass losses and soil properties (training set). The samples were collected in 2018. The second soil set of 5 grassland and 10 arable soils serves for validation of found results (validation set) and was obtained in 2019 from other sites. Importantly, the training and validation set were independent each other.





## 2.2 TG analyses and TML determination

Prior to TG analyses the air dried and 2 mm sieved soils were stored in a a desiccator to equilibrate at 43 % relative humidity (RH) for 3 weeks prior to analysis to insure comparable conditions for measurement of soils of different origins.

The correlations between TML and soil properties discussed in the introduction were obtained for air dried soils exposed to 76% relative air humidity (RH) prior to TG analysis (Kučerík et al., 2013). Recent results showed that correlations can also be observed for soils exposed to 43% RH; the RH closer to most laboratory conditions and easier to maintain (Kučerík et al.,

2020). For this reason, the thermogravimetric experiments were carried out in air enriched to 43 % RH (at 25 °C) by passing over an oversaturated solution of potassium carbonate as a reactive gas was used. Around 0.2 g of sample was transferred to an alumina sample holder placed into the thermoscale equipped with an autosampler (TA Instruments Q550, New Castle, Delaware, USA) from laboratory temperature ( 20°C) to 950°C with a heating rate of 5 °C $min^{-1}$. The flow rate of the reactive gas (air) was 100 mL $min^{-1}$. To maintain samples prior to analysis with the same humidity, the thermoscale autosampler was

modified and purged with the same air stream as the thermoscale furnace. All samples were analysed in triplicate. Exemplary records are reported in Supporting information, Figure S1.

The obtained dependences of mass loss on temperature were averaged and TMLs were obtained, i.e. in total 93 mass losses in 10°C intervals for each soil sample. In this study, the TMLs are reported with upper temperature limit as a subscript. For example, $TML_{100}$ refers to a thermal mass loss obtained between 90 and 100°C. Mass loss in larger temperature ranges are

reported with the whole temperature interval in the subscript, e.g. $TML_{200-300}$ indicating thermal mass loss between 200 and 300°C.

## 2.3 Determination of chemical and MB properties of soils

Soil organic carbon (SOC) and total N (TN) were determined by dry combustion using a 28 Series LECO analyser.

MB analyses were carried out on field-moist fresh, sieved samples (<2mm). Soil microbial biomass C and N ($C_{bio}$, $N_{bio}$)

was determined by means of the fumigation-extraction method according to (ISO 14240-2, 1997). Soil (10 g on dry basis) was weighed in three replicates and preincubated for one day at 25°C. If the natural water content was lower than 30% of water holding capacity (WHC), water was added to bring the sample to 60 % of WHC. Soil was fumigated with ethanol-free chloroform for 24 h. Fumigated and unfumigated soils were extracted with 40 mL 0.5 M $K_2SO_4$. Estimation of oxidizable C in the extracts was performed photometrically by the dichromate-oxidation method (Yakovchenko and Sikora, 1998). The

content of organic C of the non-fumigated soil ($C_{ext}$) was considered the labile fraction of soil C. Nitrogen in the extracts was oxidized to nitrate using the alkaline persulfate oxidation (Cabrera and Beare, 1993) and nitrate was measured photometrically at 210 nm (Kandeler, 1993). Coefficients kC=0.38 and kN=0.45 were used to calculate microbial biomass C and N (Joergensen, 1995).

Oxidizable C ($C_{OX}$) was estimated using wet digestion of 1 g soil sample with 5 ml 0.27 M $K_2Cr_2O_7$ solution and 7.5 ml

concentrated $H_2SO_4$ according to (ISO 14235, 1998).





Anaerobic ammonification (AMO) was estimated by incubating 5 g of saturated soil samples at 40°C (Bundy and Meisinger, 1994) for 9 days. The concentration of released $NH_4^+$-N was measured photometrically (Forster, 1995) after extraction using 1 M KCl. AMO was expressed as a net increase in released $NH_4^+ - N$ between days 2 and 9.

Soil suspension containing 5 g of soil, 20 mL borate buffer (75 mM, pH 10) and 2.5 ml of urea solution (4.8 g $L^{-1}$) was incubated 2h for estimation of urease activity (URE). At the end of incubation, 30 ml of a solution containing $Ag_2SO_4$ (100 mg $L^{-1}$) and KCl (2.5 M) was added to stop the reaction. A blank was prepared as described above but without the urea solution which was added at the end of incubation (Kandeler and Gerber, 1988). The concentration of $NH_4^+ - N$ was determined colorimetrically (Forster, 1995)

Determination of microbial respiration as production of $CO_2$ (RES) was measured using the titration method. Soils moistened to 60 % WHC were pre-incubated for six days at 22°C. Incubation of 15.0 g of soil on dry basis was done for 3 days under the same conditions in 350 ml flasks. Released $CO_2$ was trapped into a solution of 0.1 M NaOH and estimated by titration with 0.1 M HCl (ISO 16072, 2002).

Oxitop©-C measuring heads (Wissenschaftlich-TechnischeWerkstätten) were used for measurement of basal respiration ($R_B$), substrate-induced respiration ($R_S$) and growth respiration curves by means of estimation of $O_2$ consumption. Soil samples were pre-incubated (12.5 g for $R_B$ and $R_S$, 15 g for respiration curves, weights on dry basis) four days at 22°C. Samples were moistened to 60% WHC for $R_B$ and to 40% WHC for measurement of $R_S$ and respiration curves if the natural water content was lower. This allowed thorough mixing of soil samples with the substrate containing 8.42 g of glucose, 1.37 g of ammonium sulfate and 0.21 g of potassium dihydrogen phosphate. The added amount was 10 mg $g^{-1}$ for estimation of $R_S$ and for measurement of the respiration curves. Samples were incubated 96 h ($R_B$, respiration curves) and 10 h ($R_S$) in flasks (100 mL for $R_B, R_S$; 500 mL for respiration curves) and the decrease in pressure was regularly recorded, while released $CO_2$ was trapped into a solution of 2.5 M NaOH. Consumed $O_2$ was calculated using the equation of state of ideal gas (ISO 16072, 2002). The rates of $R_B$ and $R_S$ were calculated using linear regression, and the specific growth rate $\mu$ by means of non-linear regression from the relationship between cumulative $O_2$ consumption and time (Pell et al., 2005). $t_{peakmax}$ denotes the time elapsed between substrate application and the maximum rate of respiration.

Soil samples were preincubated two days at 25° C prior to estimation of a short-term nitrification activity (SNA) according to (ISO 15685, 2012). Thereafter 60 mL of a medium (pH 7.2) containing potassium phosphate buffer (1 mM), sodium chlorate (15 mM) and ammonium sulfate (3.78 mM) was added, the suspension was shaken and 5 ml of supernatant were taken after two and six hours. To stop nitrification, 5 ml of 4 M KCl was added. The suspension was centrifuged and nitrite was analysed in the supernatant using the method with sulfanilamide and N-(1-naphthyl-ethylenediamine) dihydrochloride (Forster, 1995).

## 2.4 Statistical data treatment

A two-step process was used in connecting TML/LTML's with measured soil properties.

In the first step, 93 TMLs were obtained for each soil using TG. Then, the Pearson correlation between the soil parameters (i.e. URE) and TMLs obtained in a specific temperature range (i.e. $TML_{40-50}$) were searched using the training set (either





grassland (11 samples) or arable soils (21 samples)). Then the search continued for other TMLs in the temperature range
30-600°C. If the p-value (probability) was not p ≤ 0.05 then two or more TMLs were used to find the best correlation.

In the second step multilinear regression models were developed based on the significant correlations found from step 1.
The general equation of the regression model could be express as

$$Y = A_0 + A_1 X_1 + A_2 X_2 + ... + A_n X_n \tag{1}$$

Where Y is the dependent variable, i.e. soil property, $A_0$ is intercept, $A_1 - A_n$ are regression coefficients and $X_1 - X_n$ are
independent variables i.e. mass losses. The statistical criteria used for selection or removal of variables from regressions were
based on either the significance (probability) of the F value, or the F value itself. The parameters, which did not correlate with
a TML at significance level < 5% were subjected to further analysis, in which two or more TMLs were involved.

The developed equations were verified using verification set consisting of 5 grassland and 10 arable soils. The criterion for
testing the significance of the regression model was correlation coefficient r and significance level p ≤ 0.05.

All the correlations were carried out in Excel and Statistica using a 95% confidence interval. Based on the correlations, the
regression equations were developed and verified using soils in verification sets.

## 3   Results

### 3.1   Relationships between chemical and MB parameters and mass losses (TML)

Figure1 A, B, C, and D report the correlation coefficients of a single TML (mass loss in 10°C intervals) with a soil property
of grass land soils. Figure2 reports the results of the same approach for arable soils. The coefficients are reported in the
temperature range 30-600°C, because the thermal degradation of soil organic matter occurs in this range, while above this
range generally only inorganic carbon is degraded and coefficients are much lower. In addition, the temperatures displayed on
the x axes are the upper limits of TMLs and the plot is a series of discrete points, which were connected by lines to enhance
plot´s readability. For better illustration, Figure1 also contains significance levels p ≤ 0.05, 0.01 and 0.001 (horizontal dashed
lines).

It can be seen that the individual TMLs are parts of usually wider temperature interval in which the correlation coefficients are
high. Comparison of results in both figures suggests slightly higher correlations between MB parameters and TML in arable
soils. This is also confirmed in Table 1, which reports the multilinear regressions between TML and MB with the highest
correlation coefficients and significance level ≤ 5% (Table 1). As aforementioned, the parameters, which did not correlate with
a TML at significance level < 5% were subjected to further analysis, in which two or more TMLs were involved (Equation 1).
The results are also reported in Table 1.

Table 2 then summarizes the Pearson correlation coefficients between mass losses in large thermal mass loss areas (LTML)
and soil properties. Also in this case can be seen that some LTML correlate with MB soil properties.



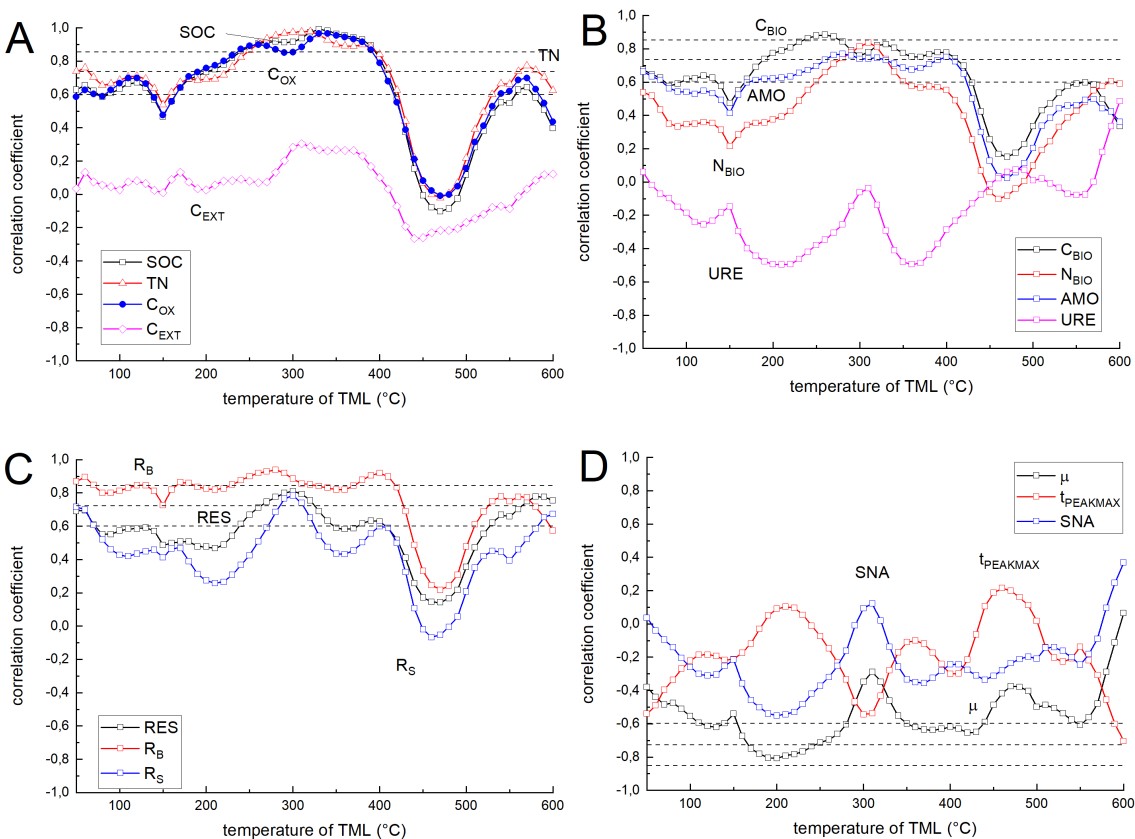

**Figure 1.** Correlation coefficients of the relationship between TMLs at different temperatures and soil properties for grassland soils. For the abbreviations see the text. The dashed lines mark the significance levels (N = 11) for p < 0.05 is r = 0.60, for p < 0.01 is r = 0.73 and for p < 0.001 is r = 0.85.

The same approach was applied to seek for the relationship between soil MB properties and LTMLs, i.e. mass losses in larger temperature areas. Table 3 reports the continuing numbering of the LTML-based equations for both grassland and arable soils, which are in Table 2 indicated in bold.

## 3.2 Verification

As aforementioned in the experimental part, additional sets of arable and grassland soils were used for verification. The verification results of the developed equations are reported in Table 4 for both TML- and LTML-based equations. As a measure of closeness between predicted and measured MB values (goodness of fit) we used the Pearson correlation coefficient, slope and intercept of the predicted and measured values. An optimal fit is Pearson coefficient and slope close to 1, and intercept close to





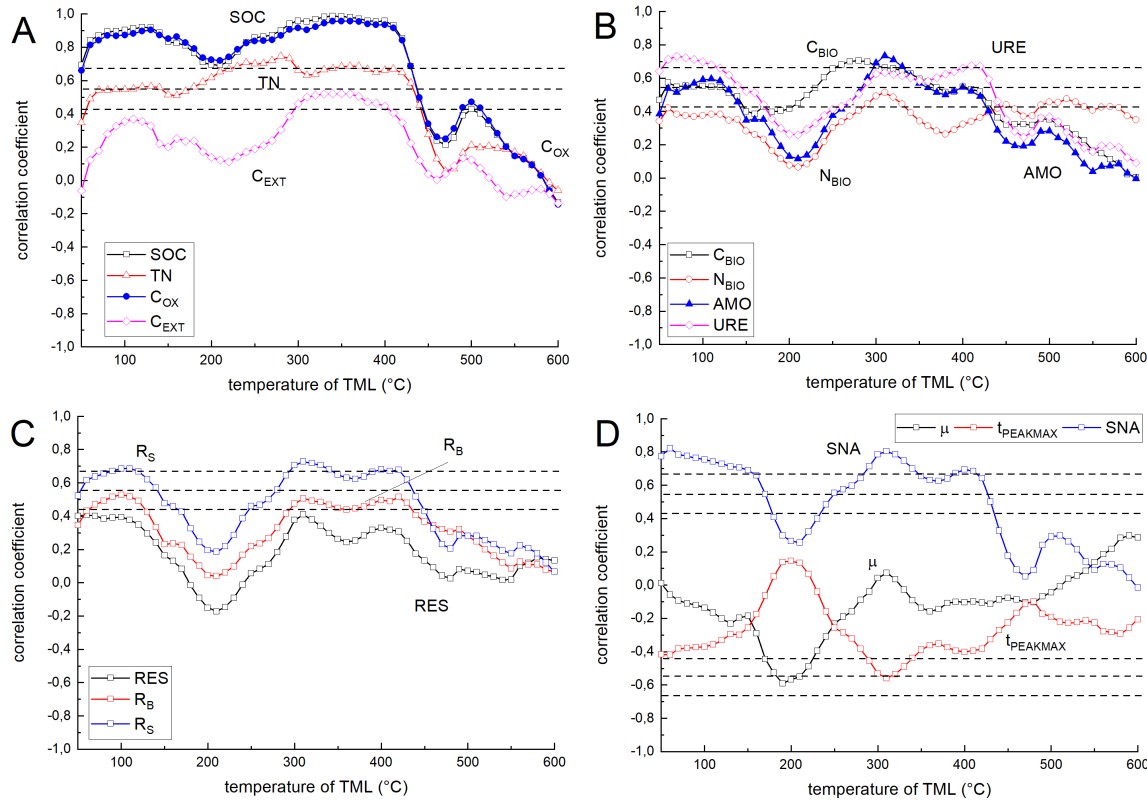

**Figure 2.** Correlation coefficients of the relationship between TMLs at different temperatures and soil properties for arable soils. For the abbreviations see the text. The dashed lines mark the significance levels (N = 21) for p < 0.05 is r = 0.43, for p < 0.01 is r = 0.55 and for p < 0.001 is r = 0.67.

zero. Reported in Table 4 are only those results, which can be considered as a positive validation of equations, i.e. correspond to the statistical criteria significance level p ≤ 0.05. The Table 4 shows that only a limited number of equations can be used for prediction of MB parameters. In grassland soils, many equations failed to predict the soil properties including TN. Also, the predictability of SOC appeared relatively weak. In arable soils, more equations seem to be applicable for prediction of soil properties including $C_{BIO}$ or URE. Also TN and SOC seem to be in arable soils better predictable.



**Table 1.** Regression models describing link between soil properties and mass losses obtained by thermogravimetry. The parameters are abbreviated as follows: Total nitrogen (TN), soil organic carbon (SOC), oxidizable Carbon ($C_{OX}$), extractable carbon ($C_{EXT}$), microbial biomass carbon ($C_{BIO}$), microbial biomass nitrogen ($N_{BIO}$), respiration - oxygen consumption (RES), basal respiration ($R_B$), substrate induced respiration ($R_S$), specific growth rate ($\mu$), time to reach maximum rate of respiration ($t_{PEAKMAX}$), anaerobic ammonification (AMO), nitrification activity (SNA), urease activity (URE)

| | | Grassland soils | | Arable soils | | |
|---|---|---|---|---|---|---|
| No. | Soil prop. | Equation | R (p) | Equation - arable | R (p) | *units* |
| 1 | **SOC** | $14.6 \times TML_{330} - 0.21$ | **0.99 (<0.01)** | $12.3 \times TML_{350} + 0.2$ | **0.99 (<0.01)** | $mg \times g^{-1}$ |
| 2 | **TN** | $1.71 \times TML_{320} - 0.06$ | **0.98 (<0.01)** | $0.89 \times TML_{290} + 0.02$ | **0.75 (0.01)** | $mg \times g^{-1}$ |
| 3 | **C$_{OX}$** | $15.7 \times TML_{330} - 0.48$ | **0.97 (<0.01)** | $11.1 \times TML_{360} + 0.3$ | **0.96 (<0.01)** | $mg \times g^{-1}$ |
| 4 | **C$_{ext}$** | $15.7 \times TML_{160} + 271 \times TML_{310} - 1146 \times TML_{410} + 56$ | **0.68 (2.13)** | $368 \times TML_{340} - 97 \times TML_{80} + 23$ | **0.74 (0.01)** | $\mu g \times g^{-1}$ |
| 5 | **C$_{BIO}$** | $1353 \times TML_{260} - 32.4$ | **0.88 (0.04)** | $906 \times TML_{290} - 19$ | **0.73 (0.02)** | $\mu g \times g^{-1}$ |
| 6 | **C$_{BIO}$** | $8.3 \times TML_{260} - 12.3 \times TML_{170} + 0.76$ | **0.94 (<0.01)** | $2966 \times TML_{280} - 2815 \times TML_{250} - 31$ | **0.78 (<0.01)** | $\mu g \times g^{-1}$ |
| 7 | **N$_{BIO}$** | $548 \times TML_{310} - 43$ | **0.84 (0.12)** | $548 \times TML_{310} - 43$ | **0.51** | $\mu g \times g^{-1}$ |
| 8 | **N$_{BIO}$** | $1229 \times TML_{320} - 917 \times TML_{370} - 43$ | **0.90 (0.02)** | $2392 \times TML_{260} - 2666 \times TML_{250} + 16$ | **0.77 (<0.01)** | $\mu g \times g^{-1}$ |
| 9 | **RES** | $2.73 \times TML_{300} + 0.42$ | **0.81 (2.51)** | # | **# (<0.01)** | $\mu g\ CO_2\text{-}C \times g^{-1} \times h^{-1}$ |
| 10 | **RES** | $6.8 \times TML_{250} - 9.8 \times TML_{220} + 0.64$ | **0.91 (0.01)** | $6.3 \times TML_{300} - 11 \times TML_{240} + 0.7$ | **0.70 (0.04)** | $\mu g\ CO_2\text{-}C \times g^{-1} \times h^{-1}$ |
| 11 | **R$_B$** | $7.72 \times TML_{280} + 0.11$ | **0.94 (<0.01)** | $15.7 \times TML_{300} - 15.1 \times TML_{270} + 0.9$ | **0.66 (0.11)** | $\mu g\ CO_2\text{-}C \times g^{-1} \times h^{-1}$ |
| 12 | **R$_S$** | $134 \times TML_{300} - 9.24$ | **0.76 (0.63)** | $130 \times TML_{320} - 3.69$ | **0.73 (0.02)** | $\mu g\ CO_2\text{-}C \times g^{-1} \times h^{-1}$ |
| 13 | **RS** | $338 \times TML_{300} - 321 \times TML_{330} - 1.56$ | **0.92 (<0.01)** | $262 \times TML_{300} - 328 \times TML_{240} + 2.5$ | **0.83 (<0.01)** | $\mu g\ CO_2\text{-}C \times g^{-1} \times h^{-1}$ |
| 14 | **$\mu$** | $-1.1 \times TML_{200} + 0.21$ | **-0.81 (2.51)** | # | **# (<0.01)** | $h^{-1}$ |
| 15 | **$\mu$** | $1.31 \times TML_{310} - 1.05 \times TML_{330} + 0.14$ | **0.92 (<0.01)** | $0.39 \times TML_{310} - 2.65 \times TML_{200} + 0.2$ | **0.83 (<0.01)** | $h^{-1}$ |
| 16 | **t$_{peakmax}$** | $-410 \times TML_{450} + 44**$ | **-0.79 (3.82)** | # | **# (<0.01)** | $h$ |
| 17 | **t$_{peakmax}$** | $238 \times TML_{340} - 338 \times TML_{310} + 41$ | **0.91 (0.010)** | $720 \times TML_{240} - 515 \times TML_{260} + 35$ | **0.88 (<0.01)** | $h$ |
| 18 | **AMO** | $34.2 \times TML_{280} - 3.75$ | **0.77 (5.57)** | $28.1 \times TML_{310} - 0.70$ | **0.73 (0.02)** | $\mu g\ NH_4^+ - N_2 \times g^{-1} \times d^{-1}$ |
| 19 | **AMO** | $239 \times TML_{410} - 203 \times TML_{150} - 5.7$ | **0.92 (<0.01)** | $61 \times TML_{300} - 85 \times TML_{240} + 1.3$ | **0.87(<0.01)** | $\mu g\ NH_4^+ - N \times g^{-1} \times d^{-1}$ |
| 20 | **SNA** | $19235 \times TML_{320} - 22757 \times TML_{350} - 110$ | **0.95 (<0.01)** | $5677 \times TML_{310} - 8262 \times TML_{220} - 108$ | **0.86 (<0.01)** | $ng\ NO_2^- - N \times g^{-1} \times h^{-1}$ |
| 21 | **URE** | $-230 \times TML_{180} + 38.4*$ | **- 0.87 (0.05)** | $225 \times TML_{410} - 11.7*$ | **0.87 (<0.01)** | $\mu g\ NH_4^+ - N \times g^{-1} \times h^{-1}$ |
| 22 | **URE** | $357 \times TML_{320} - 477 \times TML_{350} + 20$ | **0.93 (<0.01)** | # | **# (<0.01)** | $\mu g\ NH_4^+ - N \times g^{-1} \times h^{-1}$ |

# significantly better correlation was not found

## 4 Discussion

### 4.1 The correlations between mass losses and soil parameters

As it can be seen the training sample set provided correlation between soil microbiological parameters and TG data in both
soil sets. The closeness of correlation for TMLs and LTMLs is similar for arable and grass land soils with few exceptions. A
very significant difference was observed for TN, which can be explained based on work of (Trasar-Cepeda et al., 1997), who
demonstrated that relatively undisturbed soils are characterized by an equilibrium between TN and various biological activity
characteristics such as microbial biomass C, N-mineralization capacity, and the activities of phosphatase, $\beta$-glucosidase and
urease. The equilibrium can be disrupted by various chemical stresses (contamination, pH alteration) and physical disturbances
(tillage, wet–dry or freeze–thaw cycles) Chaer et al. (2009) and agricultural activity (Miguéns et al., 2007). As a result, TN is
sensitive to many management practices as fertilization, tillage and others with (perhaps temporary) reduction of correlation





**Table 2.** Pearson correlation coefficients between soil properties and mass losses in large temperature areas (LTML). The parameters are abbreviated as follows: Total nitrogen (TN), soil organic carbon (SOC), oxidizable Carbon ($C_{OX}$), extractable carbon ($C_{EXT}$), microbial biomass carbon ($C_{BIO}$), microbial biomass nitrogen ($N_{BIO}$), respiration - oxygen consumption (RES), basal respiration ($R_B$), substrate induced respiration ($R_S$), specific growth rate ($\mu$), time to reach maximum rate of respiration ($t_{PEAKMAX}$), anaerobic ammonification (AMO), nitrification activity (SNA), urease activity (URE)

| **GRASS** | **SOC** | **TN** | **$C_{OX}$** | **$C_{EXT}$** | **$C_{BIO}$** | **$N_{BIO}$** | **RES** | **$R_B$** | **$R_S$** | **$\mu$** | **$t_{peakmax}$** | **AMO** | **SNA** | **URE** |
|---|---|---|---|---|---|---|---|---|---|---|---|---|---|---|
| $LTML_{30-100}$ | 0.77 | 0.76 | 0.82 | 0.54 | 0.49 | 0.44 | 0.30 | 0.72 | 0.42 | -0.50 | -0.12 | 0.73 | -0.30 | -0.58 |
| $LTML_{100-200}$ | 0.71 | 0.64 | 0.78 | 0.59 | 0.50 | 0.35 | 0.17 | 0.76 | 0.26 | **-0.73** | 0.17 | 0.67 | **-0.58** | **-0.79** |
| $LTML_{200-300}$ | **0.91** | **0.81** | **0.92** | **0.84** | **0.75** | **0.68** | 0.39 | **0.92** | **0.55** | **-0.70** | -0.05 | **0.85** | -0.43 | -0.73 |
| $LTML_{300-450}$ | **0.89** | 0.74 | **0.92** | **0.83** | **0.61** | 0.59 | 0.30 | 0.86 | 0.52 | **-0.68** | -0.05 | **0.86** | -0.42 | -0.70 |
| $LTML_{450-550}$ | -0.10 | -0.20 | 0.04 | 0.02 | 0.17 | -0.29 | -0.19 | 0.09 | -0.54 | -0.48 | **0.77** | 0.02 | -0.47 | -0.40 |
| $LTML_{110-550}$ | 0.82 | 0.69 | 0.87 | 0.75 | 0.67 | 0.53 | 0.27 | 0.84 | 0.38 | -0.76 | 0.13 | 0.79 | -0.52 | -0.78 |
| $LTML_{200-550}$ | 0.83 | 0.69 | 0.88 | 0.76 | 0.69 | 0.55 | 0.29 | 0.84 | 0.39 | -0.75 | 0.12 | 0.79 | -0.50 | -0.76 |
| **ARABLE** | **SOC** | **TN** | **$C_{OX}$** | **$C_{EXT}$** | **$C_{BIO}$** | **$N_{BIO}$** | **RES** | **$R_B$** | **$R_S$** | **$\mu$** | **$t_{peakmax}$** | **AMO** | **SNA** | **URE** |
| $LTML_{30-100}$ | 0.53 | 0.24 | 0.55 | 0.42 | 0.40 | 0.18 | 0.19 | 0.28 | 0.45 | -0.16 | -0.26 | 0.26 | 0.58 | 0.67 |
| $LTML_{100-200}$ | 0.52 | 0.31 | 0.56 | 0.49 | 0.42 | 0.14 | 0.06 | 0.16 | 0.38 | -0.37 | -0.10 | 0.27 | 0.55 | 0.59 |
| $LTML_{200-300}$ | 0.86 | **0.79** | 0.86 | 0.33 | **0.63** | 0.43 | 0.30 | 0.29 | 0.56 | -0.31 | -0.27 | **0.66** | **0.72** | 0.66 |
| $LTML_{300-450}$ | **0.91** | 0.64 | **0.91** | **0.63** | 0.41 | **0.46** | 0.37 | **0.54** | **0.73** | -0.09 | **-0.45** | 0.54 | 0.61 | **0.85** |
| $LTML_{450-550}$ | 0.31 | 0.36 | 0.35 | 0.04 | 0.49 | 0.35 | 0.17 | 0.17 | 0.29 | -0.31 | -0.08 | 0.48 | 0.48 | 0.36 |
| $LTML_{110-550}$ | 0.85 | 0.71 | 0.87 | 0.45 | **0.62** | **0.47** | 0.32 | 0.40 | 0.64 | -0.30 | -0.32 | 0.64 | 0.73 | 0.78 |
| $LTML_{200-550}$ | 0.87 | 0.74 | 0.88 | 0.44 | 0.61 | **0.50** | 0.35 | 0.42 | 0.66 | -0.27 | -0.34 | 0.67 | 0.73 | 0.77 |

coefficients between TML and TN in arable soils. This may explain the lower correlation for arable soils comparing the grassland soils.

The existing relationship between SOC and other soil properties (Table 1) as well as the relationship of SOC with TML
(Figures 1 and 2, Table 1) may explain relationships of TML with biochemical and biological parameters. Both $C_{BIO}$ and $N_{BIO}$ were closely related to SOM content and quality (Chaer et al., 2009). Shifts in composition of SOM and microbial community could reduce their predictability in arable soils by disturbances in regulation processes by same way as for carbon and nitrogen.

The correlation between respiration and TML has already been demonstrated for soils exposed to 76% RH prior to TG analy-
sis (Kučerík et al., 2013; Siewert et al., 2012). The correlation coefficients increased with microbial respiration measurements, which enables prediction of microbiological activity using $TML_{100}$ or $TML_{300}$ (Kučerík and Siewert, 2014). The results obtained in the current study confirm the earlier conclusion about the influence of agricultural practices on this relationship (Siewert et al., 2012).

Substrate-induced respiration ($R_s$) identifies metabolically active components of the soil microbial community. Our results
show a close correlation with thermal mass losses (TML).





**Table 3.** The equations obtained from correlation between MB and LTMLs with the highest Pearson correlation coefficients as reported in Table 3. The parameters are abbreviated as follows: Total nitrogen (TN), soil organic carbon (SOC), oxidizable Carbon ($C_{OX}$), extractable carbon ($C_{EXT}$), microbial biomass carbon ($C_{BIO}$), microbial biomass nitrogen ($N_{BIO}$), basal respiration ($R_B$), substrate induced respiration ($R_S$), specific growth rate ($\mu$), time to reach maximum rate of respiration ($t_{PEAKMAX}$), anaerobic ammonification (AMO), nitrification activity (SNA), urease activity (URE)

| | | Grassland soils | Arable soils |
|---|---|---|---|
| number | Soil property | Equation | Equation |
| **23** | **SOC** | $0.78 \times LTML_{200-300} + 0.47$ | $0.86 \times LTML_{300-450} + 0.11$ |
| **24** | **TN** | $0.09 \times LTML_{200-300} + 0.06$ | $0.13 \times LTML_{200-300} + 0.008$ |
| **25** | **C$_{OX}$** | $0.78 \times LTML_{200-300} + 0.47$ | $0.76 \times LTML_{300-450} -0.17$ |
| **26** | **C$_{OX}$** | $1.12 \times LTML_{300-450} - 0.28$ | $0.83 \times LTML_{200-300} + 0.25$ |
| **27** | **C$_{EXT}$** | $17 \times LTML_{200-300} + 13$ | $12 \times TML_{300-450} + 12$ |
| **28** | **C$_{EXT}$** | $21 \times LTML_{300-450} + 18$ | # |
| **29** | **C$_{BIO}$** | $86 \times LTML_{200-300} + 146$ | $76 \times LTML_{200-300} + 44$ |
| **30** | **N$_{BIO}$** | $25 \times LTML_{200-300} + 26$ | $7.33 \times LTML_{200-550} + 8.5$ |
| **31** | **R$_B$** | $0.80 \times LTML_{200-300} + 0.58$ | $0.44 \times LTML_{300-450} + 0.4$ |
| **32** | **R$_S$** | $7.7 \times LTML_{200-300} + 8.9$ | $13 \times LTML_{300-450} - 5.9$ |
| **33** | $\mu$ | $-0.04 \times LTML_{200-300} + 0.22$ | $-0.042 \times LTML_{200-300} + 0.22$ |
| **34** | $\mu$ | $-0.22 \times LTML_{110-550} + 0.26$ | $-0.22 \times LTML_{110-550} + 0.26$ |
| **35** | **t$_{PEAKMAX}$** | $15 \times LTML_{450-550} + 15$ | $15 \times LTML_{450-550} + 15$ |
| **36** | **AMO** | $4.92 \times LTML_{300-450} - 4$ | $3.73 \times LTML_{200-300} - 1.75$ |
| **37** | **AMO** | $6.27 \times LTML_{300-450} - 6.8$ | $1.53 \times LTML_{200-550} - 2.45$ |
| **38** | **SNA** | # | $559 \times LTML_{200-300} - 463$ |
| **39** | **SNA** | # | $203 \times LTML_{110-550} - 585$ |
| **40** | **URE** | $-7.95 \times LTML_{200-300} + 38$ | $14 \times LTML_{300-450} - 13$ |

# significantly better correlation was not found

The anaerobic ammonification (AMO) reflects microbial mineralisation of organic nitrogenous compounds. If chemolithotrophs perform nitrification without using organic matter for growth, it is difficult to explain the close correlation between TML and nitrification activity (SNA) for both grassland and arable soils. We can only speculate about the indirect impact of SOM e.g. via binding of ammonium as a substrate for nitrifiers or its exchange from clays.

Urease activity (URE) is known for its sensitivity to soil management, cropping history (causing its decrease) and management practice in general, organic matter content (increases upon organic fertilization), soil depth, heavy metals, and environmental factors such as temperature and pH (Martinez-Salgado et al., 2010). Interestingly, URE showed slightly better correlations for arable soils and at different TML than grassland soils. Somewhat unclear are the contradictory results of correlation between TML and URE (mainly on grassland but also arable soils), which may be explained by site- and scale-dependent



**Table 4.** Verification of equations from Table 1 and 3. Presented are only those equations giving applicable results. The parameters are abbreviated as follows: Total nitrogen (TN), soil organic carbon (SOC), oxidizable Carbon ($C_{OX}$), extractable carbon ($C_{EXT}$), microbial biomass carbon ($C_{BIO}$), microbial biomass nitrogen ($N_{BIO}$), respiration - oxygen consumption (RES), substrate induced respiration (RS), specific growth rate ($\mu$), time to reach maximum rate of respiration ($t_{PEAKMAX}$),nitrification activity (SNA), urease activity (URE)

| TML-based | | grassland soils | | | arable soils | | |
|---|---|---|---|---|---|---|---|
| Equation No. | parameter | pearson coeffcient | slope | intercept | pearson coeffcient | slope | intercept |
| 1 | **SOC** | 0.73 | 0.46 | 5.14 | 0.98 | 0.85 | 0.13 |
| 2 | **TN** | - | - | - | 0.98 | 1.10 | -0.03 |
| 3 | **C$_{OX}$** | 0.93 | 0.46 | 1.19 | 0.97 | 0.76 | -0.19 |
| 4 | **C$_{EXT}$** | - | - | - | 0.65 | 0.65 | 4.25 |
| 5 | **C$_{BIO}$** | - | - | - | 0.73 | 1.24 | 11 |
| 6 | **C$_{BIO}$** | - | - | - | 0.82 | 1.14 | 16 |
| 8 | **N$_{BIO}$** | 0.97 | 1.16 | -13 | 0.61 | 0.35 | 18 |
| 9 | **RES** | 0.58 | 2.48 | -2.53 | - | - | - |
| 10 | **RES** | 0.80 | 5.59 | -5.61 | 0.40 | 2.34 | -1.17 |
| 14 | $\mu$ | - | - | - | 0.59 | 0.48 | 0.07 |
| 21 | **URE** | - | - | - | 0.59 | 1.72 | 2.70 |
| LTML-based equations No. | | | | | | | |
| 23 | **SOC** | 0.63 | 0.42 | 1.02 | 0.96 | 0.72 | 0.41 |
| 24 | **TN** | - | - | - | 0.98 | 1.12 | 0.00 |
| 25 | **C$_{OX}$** | 0.95 | 0.79 | 1.14 | 0.99 | 0.92 | 0.43 |
| 26 | **C$_{ox}$** | 0.91 | 0.55 | | 0.96 | 1.62 | |
| 28 | **C$_{EXT}$** | 0.63 | 1.44 | -50 | 0.55 | 0.61 | 22 |
| 29 | **C$_{BIO}$** | - | - | - | 0.48 | 1.26 | 45 |
| 32 | **R$_S$** | - | - | - | - | - | - |
| 33 | $\mu$ | - | - | - | 0.45 | 0.52 | 0.06 |
| 34 | $\mu$ | - | - | - | 0.47 | 0.05 | 0.16 |
| 35 | **t$_{PEAKMAX}$** | - | - | - | 0.38 | 0.23 | 2.38 |
| 36 | **AMO** | 0.38 | 0.46 | 27 | - | - | - |
| 38 | **SNA** | - | - | - | 0.39 | 1.13 | -454 |
| 40 | **URE** | - | - | - | 0.65 | 1.15 | 6.80 |

effect of putative soil changes on URE. We presume this may be caused (among others) by random sampling incoherence similarly as referred by (Corstanje et al., 2007), who showed that urease activity and SOC were found to be uncorrelated at shorter spatial scales ($\leq 1$ m) but significantly positively correlated at longer scales of ($\geq 15$ m).





## 4.2 The verification of results

As it could be observed in Table 4, the validation confirmed mostly the relationships for arable soils, which properties modelled
using TG correlated similarly as the training set. Better results were observed for TML – based equations than for LTML. For
the grass land soils, the verification confirmed validity only of TML-based equations for $C_{OX}$ and respiration, and surprisingly
also for $N_{BIO}$. LTML-based equations, good results were observed only for $C_{OX}$.

In our previous works, we analysed >300 untouched soils of various origin and composition sampled all over the word
e.g. (Kučerík et al., 2018; Kučerík and Siewert, 2014; Siewert and Kučerík, 2015). We have demonstrated that the TML and
LTML are useful for the determination of SOC, TN and soil organic matter fractions. In all cases, the temperatures of TMLs
correlating with SOC and TN were consistent across soil types and locations.

In the current work, we observed some significant correlations in the training soil sample set, but the validation of the results
was not always successful. This may be related to several reasons.

1. MB activity has rather the long-term than short-term effect on amount (mass loss) of soil organic matter. In other words,
the MB activity responses quickly to soil conditions by composition of soil enzymes, but the effect on SOM content is
       slower. Second issue is related to the soil sampling. The grass land soils can be considered as very stable and protected
       against physical perturbation (Jensen et al., 2019), and the correlations between TML's and soil properties are always
       stronger (Tokarski et al., 2020). However, although the grassland soils were sampled at the same time as arable soils, the
       sampling was carried out under permanent vegetation. This implies that the inputs of rhizosphere was more significant
part of grassland soils than arable soils, which influenced the validation negatively. In comparison to bulk soil, rhizosoil
       is richer in soil microorganisms, loosen separated plant cells and by roots exudates such as, organic acids, proteins and
       sugars (Hütsch et al., 2002). These rhizosphere inputs significantly influenced the results of a conventional analysis of
       soil microbiological properties.

2. The arable soils were collected in autumn, when the crop was already harvested. Therefore, the contamination by rhizo-
spheric carbon is negligible. It can be seen that predictability of these soils is better, though the number of MB parameters
       predictable by TG is also limited.

3. The factors influencing the results may also include limited and unbalanced numbers of samples both in the training set
       and in the test sets and measured values range (if a property has a relatively small range and/or low variability, it might
       be difficult to correlate as compared to a wider range up to a point).

Important is also the knowledge about the temperature range in which were the correlations observed and also the shifts
of correlating temperatures observed for arable and grass land soils. Previously, we have shown that the $LTML_{200-300}$ and
$LTML_{300-450}$ correlate with active (sum of dissolved and particulate organic matter) and intermediate (pools SOC in the sand
fraction and in stable aggregates + SOC attached to silt and clay particles). The $LTML_{450-550}$ showed a weak correlation
with a passive pool as measured by chemical oxidation (Tokarski et al., 2020). It can be observed that the higher correlation co-
efficients of microbial parameters were observed mostly in $LTML_{200-300}$ for grassland soils, but arable soils showed in some



cases shift to $LTML_{300-450}$. In accordance with previous discussion, we hypothesize that the shift can be caused by abundance of less stable fractions of SOM in grassland soils containing fresh residues and rhizospheric carbon (microorganisms, exudates etc.). l can be reached after some time (Johnston et al., 2009; Kleber and Johnson, 2010) with the amount of SOM depending on soil type and farming system (Johnston et al., 2009) and accompanied by a shift in chemical and microbiological composition of SOM and its stability (Wolińska et al., 2017).

In addition, $TML_{450-550}$ did not show any correlation with MB parameters. As this thermal interval represents the stable and inert C pool including pyrogenic C, it seems that it has no relationship to microbial activity whatsoever. This may be surprising, because microbial activity is responsible for C stabilization mechanisms including formation of aggregates (Rousk and Bengtson, 2014). Nevertheless, it implies that once C is transferred to a stabilized pool, it is unavailable for microbiological activity.

## 5    Conclusions

Mass losses recorded using TG cannot replace the traditional methods of soil MB parameters. However, in some cases TML can be a useful proxy for soil analyses and the relationship of TML to MB can be used for interpretation of TG records.

As discussed above, the limitation relates to the grassland soils, in which seems to be important the soil contamination by rhizospheric carbon comparing to arable soils. In this specific case, it is of great importance to consider the significant differences between MB activity in rhizosphere and bulk soil.

Although this work revealed links between mass losses and MB parameters, TG has limited applicability for their prediction and should be supported by other methods. In terms of carbon cycling, in the long term, the enzymatic activity determines the quality and quantity of individual pools of soil organic matter. This provides the opportunity to estimate, at least roughly, the activity of soil microorganisms. On the contrary, soil disruption causes fast changes in MB parameters, which is not immediately counterbalanced by changes in content of soil organic matter fractions. As a result, these immediate changes cannot be reflected by change in mass losses recorded by TG. Nevertheless, it remains a question, whether the deviations from these general relationships can be used for analysis of changes in SOM composition.

*Sample availability.*    The soils are a part of the regular monitoring program of the Central Institute for Supervising and Testing in Agriculture, Czech Republic. All the details, which can be published can be found in Supporting information. The rest is restricted by a Czech regulation.

*Author contributions.*    Helena Doležalová-Weissmannová (data analysis, statistical treatment); Stanislav Malý (responsible for microbiological experiments, data); Martin Brtnický (responsible for other analysis, interpretation); Jiří Holátko (microbiology expert, data assessment); Michael Scott Demyan (paper structure, explanation of some results); Christian Siewert (ideas of the paper, statistical treatment); David Tokarski (ideas of the paper, data interpretation); Eliška Kameníková (TG analysis and validation, manuscript formatting); Jiří Kučerík (ideas of the paper, paper structure).





*Competing interests.* The authors declare that they have no conflict of interest.

*Acknowledgements.* We acknowledge the financial support acquired within the LO1211, and FCH-S-21-7398 projects of the Ministry of Education, Youth and Sports of the Czech Republic. The authors thank Mr. Karel Svatoň for his excellent laboratory work.





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
