# Peer review of "Microbial soil characteristics of grassland and arable soils linked to thermogravimetry data: correlations, use and limits"

_SOIL, 2021_

## Author Comment (AC1)

This paper tries to connect soil thermal fractions with different chemical and biological properties. This connection is well referenced by literature and there are different studies reporting results about this topic. In this paper, authors change the common procedure trying to settle correlations using soil fractions from very narrow temperature ranges and they relate this narrow ranges versus total C, or total N or total microbial biomass and microbial activity. The last is difficult to understand and it is causing spurious correlations since those temperature intervals are changed based on the existence or not of the correlation. It is not clear the real goal of this work and which is the advantage of the procedure. Authors also report a high number of correlations without an interpretation of the equations obtained.

Specific comments

**Introduction**

Lines 45 to 50: It is truth that the organic mass lost by thermogravimetry can be overlapped with clay mass and carbonates depending on the clay types and clay content of samples, but evaporation occurs before the organic mass starts to combust and it can be determined by thermogravimetry as the mass lost from 50 to 180 ºC.

The temperature range of the evaporation of water from soils is also connected with a small loss of organic carbon. This mass loss is caused either by degradation (e.g. 10.1007/s10973-019-08802-8) or by sublimation (e.g. 10.1007/s00374-010-0442-3). This misunderstanding comes probably from the traditional way of TG data evaluation, which is based on using the derivative of the TG curve (DTG). The DTG minima seems to show separate temperature ranges for these processes (e.g. water loss vs. organic carbon), but the use of other techniques (EGA and others) reveals there is a small overlap.

Lines 50-51: only to try to separate CO2 and water from clays and organic samples…….which is not possible even by those methods because the CO2 from clay and organic matter overlaps from 200 to 650 ºC. This is significant for soils with low organic C content but not for soil with high C content where the contribution of clay masses is very low. I do not think they can be argued as limitations in the superficial way that is done by authors.

We are not sure if we understand well the comment of the referee. Lines 50-51 refer to the limitation of TG to consistently provide information on soil composition. In pure materials, such as oxalates and carbonates (used as TG standards), the mass losses correspond stoichiometrically to composition. From the material point of view, soil is a heterogeneous and an anisotropic material with distribution of pores, aggregates, and organic functional groups of various stability. In soil, stability of a chemical compound can have a different thermal stability as it is bound by different forces. Therefore, mass losses obtained during TG of soil do not always reflect one process. Instead, the processes are overlapping. In addition, what we see is mass loss, not behaviour of individual molecules. As a result, physical meaning of mass losses does not always correspond to biogeochemical meaning (i.e. mass loss cannot be clearly connected with degradation of soil organisms). Use of evolved gas analysis as a compliment improves the information value of the experiment.

Line 59: Check the sentence after the references. It is the term "vary" correct there?

It is a typo, it should be "various".

**Experimental**

2.2 TG analysis and TML determination

Lines 85-86: Considering that samples are combusted through the temperature scan, and that water is lost only during the first 180 ºC (excepting adsorbed water in clays) and can be easily measured, what is the reason for the procedure exposed dealing with RH?

This is probably a weakness of our introduction. In the original paper of Siewert (2004; DOI10.2136/sssaj2004.1656 ), where this approach was used for the first time, is reported the correlation of mass losses with SOC, TN, clay content and carbonates. In later works, we discussed (paper of Siewert and Kucerik) the necessity to expose the soil to constant relative humidity due to comparable conditions prior the analysis. Each soil has its capacity to adsorb some humidity when exposed to relative humidity. The final humidity of each soil is different but it reflects the properties of soil structure and enables the determination of SOC, TN etc. Without the equilibration, the soils would have different starting moisture contents which would affect the determined mass losses (determined as mass loss in a temperature interval divided by total mass including moisture). Consequently, this would affect the determined SOC, TN etc…

Lines 88-90: Most of studies using TG for soils report air flows of 50 ml/ min and temperature rates of 10 ºC / min. There is literature showing how these rates may change the evolution of the DTG curves. Is there any reason for changing those rates to 100 ml/min and 5 ºC / min?.  Specifically, too fast air flow rates can limit the complete oxidation of the organic matter.

Yes, the reviewer is right. A heating rate of 5°C/min is used due to the poor thermal conductivity of soils and higher sample mass we use in our experiments. Also, it is used to be able to compare all our results. Indeed, we use 5°C/min in all our works.

Concerning the flow rate, it is higher than in other TG works because we use higher sample mass and an excess of oxygen is need mainly at temperatures above 200°C, when the main combustion process starts is to avoid any charring or imcomplete combustion?. In fact, in our work we use two different systems, Mettler Toledo and TA Instruments and each device is specific. Mainly, MT can accommodate large sample mass (even around 1g), in this case we used even 200 ml/min. The TAI used a smaller soil sample (up to 200 mg), thus to obtain comparable conditions, we use 100 ml/min. This was extensively tested in past (unpublished results).

Lines 92-93: I do not know what you mean as "the obtained dependences of mass loss on temperature were averaged". Do you mean the soil organic matter was fractionated for different temperature intervals and shown as the average of the three replicates done? What you write is not understandable.

Yes, each sample was measured in triplicate and shown as the average of the three replicates. This sentence will be rewritten to be clearer.

2.3 Determination of chemical and MB properties of soils.  What is MB here? Why do you symbolize Microbial soil properties as MB? Would not it mean Microbial Biomass, MB ?

"MB" stands for "microbial" We have defined this in the sentence.

Lines 125-127: Why the water content change from 60 % of WHC for RB to 40 % of WHC for Rs? Substrate induced respiration adding glucose depends on water content as basal respiration.

Preliminary experiments showed that soils with higher content of clay became sticky when they were mixed with the substrate at 60% WHC. It resulted in lowering of gas exchange, i.d. in significant decrease of RS. Experiments with soils having different clay content performed at different %WHC showed that 40% WHC was an optimal value for RS for our set of soils. Similar experiments with RB confirmed generally recommended value 60%WHC for RB.

2.4 Statistical data treatment

What is TML/LTML´s ?

TML stands Thermal Mass Loss (line 56 and 93); LTML stands for large TML (there inconsistences in manuscript, we are sorry for that).

What is the sense of searching for correlation with TMLs for such a low interval of temperature, 10 ºC? What is the connection of a 10 ºC soil organic matter fraction with any of the mentioned properties? To me, that criterion may yield spurious correlations. In special if you use as a criterion to increase the temperature interval  when there is not a correlation with the 10 ºC interval until you find the correlation.

The explanation of using 10°C was reported in our papers several times and it is also discussed in our manuscript. Soils from various sources differ in their composition and give even different number of peaks when derivative TG is used. This prevents any reasonable comparison of TG records of different soils. Separation into predefined temperature intervals of soil measured between 30-950°C gives the same number of mass losses for each soil. Also, this solves the problems with overlapping processes. If for example only 1°C interval is used, then the mass losses from repeated measurements sometimes differ due to noise. Use of 10°C interval enables to decrease this noise as the reproducibility significantly increases.

As comes from previous comments, the obtained mass losses in 10°C have no biogeochemical meaning and we use them as "TG indicators". We try to find if there is a correlation between these indicators and soil properties. There are many overlapping processes occurring during soil heating and combustion and to find their meaning is impossible. For example, even when the derivative TG is used and mass losses imposed by minima cannot be interpreted in biogeochemical sense. SOM contains thousands of different molecules, some are protected by aggregates, some by interaction with minerals and this changes the temperature intervals in which occur their thermal degradation. In addition, clay minerals release the chemically adsorbed water andsome minerals are decomposed. Despite that, some of the mass losses in 10°C intervals serve as reliable indicator for SOC determination. In other words, we used TG as a fractionation technique (that fractionate soil based on thermal stability) without knowing the biogeochemical or physical meaning of obtained fractions. Searching for correlation of these fractions with soil properties is the only way to understand, at least partially, their meaning.

Then, how you can compare two sets of independent samples that have "different number of samples"? 11 grasslands versus 5 grasslands, and 21 arable samples versus 10? That is against the comparative criteria settled by statistics.

The aim was to first create or find the best regression equation and then test it on an independent set of samples. The number of samples came from the regular monitoring program of the Central Institute for Supervising and Testing in Agriculture, Czech Republic (Poláková et al., 2017). Comparisons of results is based on R and p value, which a common approach.

I do not think the statistical design be correct.

It is the same approach we used in past and reviewers had no objections. (e.g. DOI: 10.1111/ejss.12877; 10.1016/j.geoderma.2017.12.001; 10.1016/j.geoderma.2019.114124) |Therefore, we use the same approach to have consistent and comparable results. Nevertheless, we understand that for example biostatics can use different approaches to this subject.

**Results**

Figure 1: Do you represent the same SOC of one sample versus the 94 different TMLs? What is the sense of this method? What is the advantage to show results by this way? From my perspective it results very confusing and difficult to interpret. Which is the meaning of the negative correlations observed for some of the parameters?

Yes, we calculated correlation coefficients of all TMLs with respective SOC. We wanted to show in which temperature interval there are correlations of the parameter, in this case SOC, with mass loss. We choose this way of presentation to show how correlation changed with temperature and that is not an accidental correlation but there is always a connection with surrounding temperature.

The negative correlations are surprising also for us as it means that lower mass loss (i.e. lower amount of SOC of some quality) would implicate e.g. higher content of specific enzyme.

How can you explain the high correlation for RS values from 300 to 450 ºC if you added glucose? Priming effect? Is not the glucose added consumed but the C soil?

It may not have been clear that we analysed the soils without addition of glucose. Glucose was used only in the substrate induced respiration (SIR) experiments. Data from those experiments was then correlated with TML data of soils which were not amended with glucose. Between 300-450°C are mass losses correlating with active pool of SOC (DOI: 10.1016/j.geoderma.2019.114124). That is the probable explanation of this correlation.

Line 172-173: Which are the criteria to select LTMLs? In fact, the fractions would be the ones settled for the labile and recalcitrant organic matter which is something very well known.

Criteria of selection came from our previous works, in particular from paper DOI: 10.1007/s10973-014-4256-7. Those are LMLS, which resulted from mutual correlations of TML and which enabled to separate the LTMLs temeperatuer areas used in this work.

Which the usefulness or advantage of Table 1?

As already mentioned, we were searching for the correlations between TML and soil properties, those we found are listed in Table 1 in order to show the mathematical relationships and statistical relevance between those parameters . The relationships were then subjected to verification.

**Discussion**

Authors can not explain most of the results obtained excepting the common ones linked to chemical soil properties.

In light of above discussion, mass losses TML are only fractions or indicators without biogeochemical meaning. We know roughly what happens in particular temperature areas, but we can only speculate about the explanations. It is specifically mentioned in the paper (and also in our previous papers) that we are searching for correlations between TML (or LMTL) and soil properties and their possible application. These TML can then act as proxies for different soil properties for estimation.

Finding the explanations was not the aim of the paper; we even doubt that there is a simple explanation for the correlations of specific TML with soil properties.

Arguments exposed for the differences of TN among grassland and arable lands are speculative. Lower correlation simply would involve less organic N since it is not as attached to the mass lost from 200 to 450 ºC as in grasslands. The content of inorganic C, clays and carbonates of the samples could be influencing also the results.

Does this mean the measured TN or the TML correlations? We are sorry but the comment is not clear. Concerning the last sentence, inorganic C is always a very low proportion of total N, unless there has just been an application of N fertilizer or something like that. Based on the suggesiton we speculate that it would have something to do with differences in microbially processed N, POM associated N, and maybe even „black" N.

Lines 205-206: what do you mean as "prediction of microbial activity" by the TML? In special  by TML100, the fraction where evaporation starts and volatiles taking  part of the organic matter are lost.

The aim of the paper to find corelations between TML and soil properties and to obtain the equations connecting TML with soil properties obtain. They are reported in Table 1. By "prediction of microbial activity" by the TML is meant to use the TML to determine (or estimate) the microbial activity by measuring the TML and using respective equations in Table 1.

Table 3: As an example, the first equation shows the highest correlation with SOC at 200-300 ºC for grasslands and at 300-450 for arable lands. Do you really think we must use that equation to calculate SOC from those intervals? What is the really meaning and advantage of those equations given for such a narrow range of temperature? What is the meaning of the slope , SOC per degree of temperature? Or is that most of the soil C is

lost so fast from 200 to 300 ºC? What is the meaning of the ordinate, the A value of the straight line?

SOC is connected with total SOM, but the link is not straightforward and depends on many factors including land use, as suggested by our results. The mass loss between 105-550°C is traditionally used for determination of SOM (i.e. loss-on-ignition), which is then recalculated by a factor 1.724. However, according to some authors the factor can significantly vary depending on land management (10.1016/j.geoderma.2010.02.003 and references therein). Land management influences SOM quality, which is reflected in its thermal stability. For this reason, there may be differences in the temperature interval reflecting the SOC. This lends evidence as opposed to using a generic factor or conversion of SOM to SOC, management and soil specific factors alter this relationship, thus the use of more narrow TML for estimating SOC/SOM.

The interpretation of slope´s and A value meanings is not easy as the equation describes a correlation but does not imply causality.

Table 4: That is only for the temperature interval given in Table 4? What is the criterion to settle the applicability?

Table for reports the verification results of equations reported in Table 3. It refers to the equation numbers reported in Table 3. In Table 3 we report the statistical results of validation, in case of application of the equations, the results help to estimate the validity of calculated values.

With respect to Cbio: Can we consider calculating the soil microbial biomass by the equations in table 3? Both are quite similar with the exception of the A value. What about the difference?

As discussed above, the quality of SOM is related to soil quality and management. The differences in equations reflect these differences.

Lines 260-265:  It follows the same trend of the carbon. Why the correlation is lower with most of the parameters you use after 400 ºC? SOM percentages obtained by TG from 180 to 600ºC usually correlate well with total C and organic C in literature. That is the correct way to settle the correlation since what you measure is the total C and N in soil. Your procedure makes sense if you could obtain the C for the same temperature intervals by the elemental analysis.

The microbial stability of SOC roughly correlates with thermal stability, although it cannot be taken quantitatively, i.e. mass loss in some interval does no equal the amount of e.g. Cbio. Instead, the mass losses are indicators, due to overlapping processes discussed above. As the microbial parameters are related to active and labile SOC, which correlates with lower temperatures, the correlations above 400°C are weaker.

Regarding the last sentence, We used elemental analysis as the reference method for SOC and was used in correlations with TG data.

Conclusions

First paragraph: This paragraph is confusing because of the vague definition of MB commented before. TG is an useful technique to calculate soil organic matter, SOM, and

there are different references about correlations of the thermal SOM fractions given by the TG with soil elemental properties and even with soil microbial metabolism.

We generally agree, there are some works reporting connection between microbial soil data with thermal properties, but they are based on calorimetry measurements, not on the TG data. We agree that TG is a useful technique to calculate SOM content; concerning the fractions, the situation is more complicated – the fractions obtained using TG are equal to fractions obtained by procedures applied in soil sciences. Nevertheless, the TG fractions correlate with the fractions obtained by procedures generally accepted in soils sciences. Our recent paper is devoted to this subject (DOI: 10.1016/j.geoderma.2019.114124). In other worlds, using TG we apply a material science approach to determine thermal stability of a complex material, and the results may not always be clear to soil scientists, who may understand the term soil stability differently. This is a general challenge when the TG is used in soil science (and our long-term task) to connect the data obtained using TG with classical soil analyses and definitions. In this way, the use of TG in soil science can be widen and provide reasonable data.

Lines 269 to 271: You have to check that in your paper. There is not experimental evidence in your paper for that conclusion.

That is true, there is no experimental evidence, but is the hypothesis to explain the observations. We agree that the part can be better explained.

---

## Author Comment (AC2)

The authors present a manuscript where they attempt to connect incremental thermal mass loss (TML) to various metrics associated with soil quality indicators (SQI), soil health, and soil microbial activity. Standard protocols for assessing SQI typically require multiple subsamples that are prepared for different measurements at different moisture contents and narrow temperature ranges. Authors suggest that TML may be a feasible technique to acquire data for multiple SQI metrics with a single measurement by correlating TML to select SQI. The TML temperature ranges are compared to measurements of SQI and linear regression is used to create models that are predictive of SQI values based on TML measurements. Although the authors present an interesting case for investigating connections between TML and SQI, their analytical approach does not clearly answer their objective due to obscure correlations that are not clear in interpretation. The predictive equations generated from their modeled data do not seem to provide a more reliable method of interpreting SQI and the authors fail to make a case for why they believe the generated equations have merit for SQI assessment. A different approach to analysis is suggested, and if authors do not find an analysis that is more fitting to the objective, perhaps a different experimental design is also needed.

18 – SQI are not officially standardized into groups or arranged in any official capacity. Authors should mention that the SQI listed here are the ones that they have considered and that the listed parameters do not cover all SQI that could be measured

We agree with the comment, we can extend the statement.

22 – physical, chemical, and biological soil properties can change because of slight or major soil modification. I suggest avoiding categorizing them in this way because it limits which SQI are chosen to represent different soil processes.

Thank you for suggestion.

28 – What do the authors imply here by 'number of methods'. Do you refer to different methods that measure the same property or different methods to measure different SQI?

It was meant "different methods to measure different SQI".

45 – the authors state that mass losses using TG do not have a clear meaning unless connected to accessory information. This is partially accurate, but there are many experiments connecting TG measurements to accessory measurements in ways that greatly increase the ability for TG to be predictive of certain soil properties. Thinking specifically of how gravimetric water content is measured and how the C:N ratio to soil organic matter is measured. The authors suggest that fractionated TML may also be useful for assessing SQI but fail to reasonably address that existing methods are conducted at narrow temperature ranges because those narrow ranges are typically

most associated with the property being measured. How does a single measurement of TML over all those ranges collect valuable information

We agree, that in some cases TG can provide a valuable information. We feel, we should better introduce the TG technique and meaning of TML.

85 – this step to reach the same relative humidity across all samples seems unnecessary. Many researchers would instead focus on reaching a constant dry mass before analysis

Yes, we agree.

In the original paper of Siewert (2004; DOI10.2136/sssaj2004.1656 ), where this approach was used for the first time, is reported the correlation of mass losses with SOC, TN, clay content and carbonates. In later works, we discussed (paper of Siewert and Kucerik) the necessity to expose the soil to constant relative humidity due to comparable conditions prior the analysis. Each soil has its capacity to adsorb some humidity when exposed to relative humidity. The final humidity of each soil is different but it reflects the properties of soil structure and enables the determination of SOC, TN etc. Without the equilibration, the soils would have different starting moisture contents which would affect the determined mass losses (determined as mass loss in a temperature interval divided by total mass including moisture). Consequently, this would affect the determined SOC, TN etc…

88 – what is the rationale for this heating rate? A heating rate of 5*C per minute from 20-950*C minutes is approximately 186 minutes of heating on a 0.2 g sample. Do the authors have a reason for this protocol and why they expected this to produce reasonable results? There is no reference mentioned in this section

Heating rate 5°C/min is based on long-term experience with use of TG in soil analysis and it started by the first paper on this "series", Siewert 2004. Since this time we use it in our experiments and it allows us to compare all our results.

From the material science point of view, this heating rate is optimal also due to poor thermal conductivity of analysed soils, conditions of oxidation (air flow rate) and higher sample mass.

Importantly, in our works we do not use ground soil, which may cause a problem with reproducibility. Nevertheless, 0.2 is still enough to have good reproducibility of measurements without grinding and soil homogenization. It should be noted that 0.2 g is quite large sample mass, as in TG smaller sample masses are used. In addition, even the thermobalances providers prefer to decrease sample mass and increase the sensitivity, thus the commercially available devices can rarely use this mass.

In fact, in our work we use two different systems, Mettler Toledo and TA Instruments and each device is specific. Mainly, MT can accommodate large sample mass (up to 1 g), in this case we use an oxygen flow of 200 ml/min. The TAI used a lower samples mass, and to obtain comparable conditions, we use an oxygen flow of100 ml/min. This was extensively tested in past (unpublished results).

We agree with the reviewer and have added method references.

102 – what is meant by water holding capacity compared to water content? I interpret this to mean the water holding capacity of an intact soil sample based on porosity, texture, and related factors. Was WHC measured on these soils based on their natural state before being disturbed?

Yes, the interpretation of the reviewer is correct, it is derived from WHC measured in samples before being disturbed.

141 – at this point in the manuscript, the term LTML has not been described. I think it should not be abbreviated here.

Yes, it is erroneously described on the line 94 and 95.

144 – how are the TML being correlated to soil parameters here. Is the same TML range for each soil sample being correlated to the soil parameter measurement for each soil sample? It seems like this is what is being done, but please elaborate more clearly for readers.

Yes it is as stated by the reviewer. We have corrected it in the text.

160 – Although people highly versed in the field may know this information, it is important to include citations about the 30-600*C temperature range you are referring to for SOM degradation.

Yes, we agree and have added in a reference.

169 – What is the meaning of the equations when two or more TML ranges are used. How are we to interpret the meaning of each variable attached to this equation?

The TML have thus far no direct biogeochemical or physical meaning (although it has physically meaningful unit), they can be considered as indicators, thermal fractions, or general proxies with no clear meaning. The aim of the work was to find possible links, i.e. correlations, between TML or LTML with soil properties. Thus, we combined either one or more TML to see whether this correlation exists. If exists, then, if it is possible to use it for prediction of soil properties. Importantly, this correlation does not imply causality.

171 – Does your selection of large thermal mass loss areas have a significant quantitative meaning? It seems that you have selected wide ranges but do not explain a meaning for each lower and upper limit. This is also important because LTML values from table 2 are used to determine which linear equations are appropriate for further discussion in table 3 and beyond.

Criteria of selection came from our previous works, in particular from paper DOI: 10.1007/s10973-014-4256-7. For soils from all over the world, regardless the type and origin, the mutual correlations of TML resulted in regions of temperatures, where the TMLs mutually correlated; those areas are limits of LMTLs. In particular, we observed the areas up to 100°C, 100-200, 200-300, 300-450, 450-550°C and others. In other words, these are TG fractions, which are distinguished from others by mutual relationships of their components. These TG fractions correlate with classical soil fractionation techniques (10.1016/j.geoderma.2019.114124), e.g. with fractions obtained by approach

suggested by Zimmermann et al. ( 10.1111/j.1365-2389.2006.00855.x). Also, their correlation with C, N and clay is close but not straightforward (10.1016/j.geoderma.2017.12.001).

179 – for table 4, are there fewer applicable results for grassland because grassland had a smaller sample size? This outcome should have more explanation.

No, but the reviewer is right, this should be better explained.

190 – you state that the closeness between TML and LTML correlation is close with a few exceptions. Is there interpretation about why some correlations were not close and others were (other than TN, for which you do provide speculation)? Does it have something to do with the LTML ranges selected for correlation? Other factors?

As aforementioned, TMLs (and also LTMLs) have no clear meaning and represent indicators or thermal fractions of soil. Soil organic matter, which is the part of soil degraded within 200 to around 550°C, is a continuum of large amount of molecules separated based on their thermal stability. One molecule or group of molecules, can be due to various stabilization mechanisms degraded in various TMLs. For this reason, the choose of LTML range may not be that important as it seems to be.

The aim of the paper was not to search for reasons of correlations, we provided only several hypothesis in cases we considered most important and logical. The aim was to find if the correlations exist, how can be mathematically described and whether are applicable. What we found that there is a link in some cases, but the applicability is problematic.

193 – Although there is speculation about why TN was among the biggest differences between the two soil types, the authors neglect to mention the relevant temperature ranges for soil N and why correlations with TML outside of those ranges would have meaning in this measurement. Are the authors confident that N is a significant part of mass loss across the entire range specified?

An interesting question. The manuscript needs better explain the specifics of TG measurement and meaning TMLs/LTMLs. An important assumption is the close link between C and N; they are linked in biogeochemical cycles and in various forms of N. Yes, we are confident that N is a part of mass loss across the entire range. In fact, mass loss in TG cannot be understood in terms of spectroscopy, where functional groups have some superposition; instead, it is a continuous mass loss caused by degradation or transformation (char formation) reactions. But was a very valuable comment.

200 - It is well known that microbial biomass C and N are correlated with SOM, but your interpretation does not explain why TML in different temperature ranges are useful for this interpretation. For example, many researchers measure SOM by combustion between 300-400*C. Why are measurements outside of this range also valuable? Please elaborate.

This is a valuable suggestion, we will elaborate it.

205 – Belaboring the point here, but this is important for discussion. Microbial respiration in soil and microbial activity above 100°C is unlikely to have much meaning in practical situations. A measurement above 200°C is unlikely to be predictive of any microbial activity unless the prediction is that there is little to no microbial activity. The vast majority of microbes and microbial exudates are not part of the active C fraction at this point and greater. What do these correlations mean?

We agree, this point should be better explained and connected with the discussion about the correlation between TML and respiration.

211-221 – Similar criticisms toward interpretation of N compounds. The authors present speculation with little connection to the objective based on TML and its use to interpret and assess results for different SQI

We agree, better connection to the objective based on TML should be presented.

234 – I would like to see more exploration about how these factors like MB, TN, SOC, etc. overlap in terms of TML within a certain range. Considering most of the temperatures in the incremental TML are outside of microbial activity range of soil, I am curious to know if the correlations are confounded by other factors that are not currently discussed in the manuscript. The authors should discuss this in order to make their argument for using this method more convincing.

That is an interesting suggestion, this may improve the paper.

239 – I think your data do not currently support the idea that rhizosphere inputs for grassland are what negatively affected the validation. As stated on line 247, the sample set is limited and unbalanced. Authors are far too speculative in this regard.

The influence of rhizosphere was the hypothesis of the reasons of observed inconsistencies. We are aware that this hypothesis is not supported by the experiments, it is more a logical deduction. We wanted, among others, to stress out the problems when using thermogravimetry for prediction of microbial data that the sampling under various vegetation cover is in this case an important issue. Yes, the number of samples is limited by the project, but it is still indicative to draw a hypothesis. Perhaps, we should reconsider how to present our hypothesis as a one of many others.

262 – Microbial activity can still be correlated with stable C fractions. This data has been observed. I am not confident that authors have shown that the thermal intervals measured in this way are associated with microbial activity. It would be interesting to see how the measured microbial and SOM parameters correlated to each other rather than the TML.

Indeed, they correlate, we did not show the data, as it would disturb the red line of the paper – finding correlations and their verification. But the reviewer is right, it may

help to improve the understanding the meaning of correlations, we may include it in a supplement.

268 – TML may be a useful proxy for some soil analyses, but the way that authors have analyzed data in this manuscript does not show this. Interpretations in this manuscript drifted away from the proposed objective of showing how TML is connected to various SQI. Authors present very little data and interpretations that answer this question in a coherent way.

As aforementioned, the aim was not to interpret the found correlations, as for that we would need additional analyses. Nevertheless, based on both reviewers suggestions, in some cases, the interpretation can be improved.

267 – authors make claims in this concluding paragraph that are not supported by their data and interpretations. TML does not appear to be a useful proxy for the soil analyses mentioned because authors did not present a strong case for a reliable or more convenient predictive model. The validation step failed in most cases for grassland soil and interpretations of the model for arable soil are not well supported in the manuscript. Authors may benefit from adjusting the overall objective and analysis methods so that the value of TML data is more apparent to readers, specifically for matters of SOM and its various fractions. The TML connection to microbial activity is likely confounded by chemical fractions of SOM that authors did not do a satisfactory job of parsing through in their results and interpretations.

We agree, there are issues in the manuscript that can be improved based on the suggestions of reviewers.

Technical error

59 – the word 'vary' may be a typo with the intended word as 'various'.

We have correct this misspelling.

252 – I think the intention is to write 'intermediate pools (…' rather than 'intermediate (pools…' Parenthesis after the word 'pools'.

The reviewer is right, this has been corrected.

Figures 1 and 2 should include the full text of abbreviated terms in the description (e.g. SOC = soil organic carbon).

We have added in the description of these abbreviations.